# Are behavioral economics interventions effective in increasing colorectal cancer screening uptake: A systematic review of evidence and meta-analysis?

**Bahman Ahadinezhad**[1], **Aisa Maleki**[2], **Amirali Akhondi**[2], **Mohammadjavad Kazemi**[2], **Sama Yousefy**[2], **Fatemeh Rezaei**[2], **Omid Khosravizadeh**[1] *

**1** Social Determinants of Health Research Center, Research Institute for Prevention of Non-Communicable Diseases, Qazvin University of Medical Sciences, Qazvin, Iran, **2** Student Research Committee, Qazvin University of Medical Sciences, Qazvin, Iran

* Omid.khosravizadeh@gmail.com

**Data Availability Statement:** All relevant data are within the paper and its Supporting Information files.

## Abstract

Various interventions have been investigated to improve the uptake of colorectal cancer screening. In this paper, the authors have attempted to provide a pooled estimate of the effect size of the BE interventions running a systematic review based meta-analysis. In this study, all the published literatures between 2000 and 2022 have been reviewed. Searches were performed in PubMed, Scopus and Cochrane databases. The main outcome was the demanding the one of the colorectal cancer screening tests. The quality assessment was done by two people so that each person evaluated the studies separately and independently based on the individual participant data the modified Jadad scale. Pooled effect size (odds ratio) was estimated using random effects model at 95% confidence interval. Galbraith, Forrest and Funnel plots were used in data analysis. Publication bias was also investigated through Egger's test. All the analysis was done in STATA 15. From the initial 1966 records, 38 were included in the final analysis in which 72612 cases and 71493 controls have been studied. About 72% have been conducted in the USA. The heterogeneity of the studies was high based on the variation in OR ($I^2 = 94.6\%$, heterogeneity $X^2 = 670.01$ (d.f. = 36), $p < 0.01$). The random effect pooled odds ratio (POR) of behavioral economics (BE) interventions was calculated as 1.26 (95% CI: 1.26 to 1.43). The bias coefficient is noteworthy (3.15) and statistically significant ($p < 0.01$). According to the results of this meta-analysis, health policy and decision makers can improve the efficiency and cost effectiveness of policies to control this type of cancer by using various behavioral economics interventions. It's noteworthy that due to the impossibility of categorizing behavioral economics interventions; we could not perform by group analysis.

## Introduction

Colorectal cancer is the third leading cause of death in the world, and it is expected in 2021, about 52,980 people in the United States will die from colon cancer [1]. In 2020, colorectal cancer accounted for 935,173 deaths [2]. In 2019, colorectal cancer was the second leading cause

**Funding:** The author(s) received no specific funding for this work.

**Competing interests:** The authors have declared that no competing interests exist.

of disability-adjusted life years (DALYs) [3]. Between 1990 and 2019, worldwide, colorectal cancer cases more than doubled. So that cases have increased from 842098 to 2.17 million and deaths have increased from 518,126 to 1.09 million [4]. During this period, a significant increase in incidence is observed in adults under 50 years of age, especially in countries with high socio-demographic index [4].

Globally, behavioral factors were the main drivers of DALYs from colorectal cancer in 2019 [4]. Colorectal cancer is most commonly diagnosed in people aged 65 to 74 years, but it is estimated that 10.5 percent of new cases of colorectal cancer occur in people younger than 50 years [5]. According to the recommendations of the World Health Organization and the American Cancer Prevention Association, all people over the age of 50 are at risk of developing colorectal cancer and should accomplish various screening methods [6].

Screening can be effective in reducing the incidence and mortality rate of colorectal cancer [7]. The US Preventive Services Task Force has concluded with high confidence that colorectal cancer screening in adults aged 50 to 75 years has a significant net benefit [5]. Also, this group has concluded with moderate certainty that colorectal cancer screening in adults aged 45 to 49 years has a moderate benefit [5].

Numerous interventions such as advance notification letters, postal reminders and phone navigation calls have been employed to improve the uptake of CRC screening [8]. Through a scoping review, Leach et al. [9] investigated the impact of various interventions on colorectal cancer screening in the United States. They reviewed 13 projects promoting CRCS. The obtained results showed that in the intervention arms, performing colorectal cancer screening by any modality was 16% more than the control group. The result gained from the review of the evidence related to low-income and middle-income countries disclosed that the interventions had a positive effect on increasing the uptake of colorectal cancer screening [10]. Conclusion of another review confirmed the beneficial effect of removing financial obstacles in increasing colorectal cancer screenings [11].

Recently, BE based interventions have been implemented by researchers. Such interventions seek to utilize the heuristics employed by human in making judgments and decisions [12–14]. Designed interventions usually involve small changes and nudge people [15]. Different types of nudging tools can be used to improve the uptake of CRC screening [13]. Choice architecture, perceived social norms, default options, financial incentives, and other techniques can all be applied to influence screening behaviors [13]. Huf et al. [16], Becker et al. [17], Lieberman et al. [18], Green et al. [19], Nisa et al. [20], Slater et al. [21], Mehta et al. [22], Mehta et al. [23], Ritvo et al. [24], Kullgren et al. [25], Vanroosbroeck et al. [26] have confirmed the positive results of behavioral economic interventions. Studies using behavioral economics interventions to promote colorectal screening have reported heterogeneous results. In a rapid review, Taylor et al. [8] concluded that behavioral economics interventions have a mixed effect on increasing the uptake of colorectal cancer screening. To date, the robustness of empirical and confirmatory evidence regarding the effectiveness of BE interventions in promoting the use of CRC screening remains unclear. Therefore, in this paper, the authors have attempted to provide a pooled estimate of the effect size of the BE interventions running a systematic review based meta-analysis.

## Materials and methods

### Search strategy

This research was accomplished based on PRISMA guidelines. Searches were performed in PubMed, Scopus and Cochrane databases. The keywords used in the search are: Bowel cancer, Colorectal cancer, Faecal immunochemical test, Faecal occult blood test, Colonoscopy,

Flexible Sigmoidoscopy, Colonography, Screen, Behavioral economics, Nudge, Incentive, Norms, Default, Salience, Priming, Commitment, Heuristics, bias, aversion, decision fatigue, regret, Order effect, Behavior, Participation, Adherence, Uptake, Utilization, Practices. Search strategies according to databases are presented in Table A in S1 Appendix.

### Eligibility criteria

We included studies that met the following eligibility criteria: 1) had a randomized trial design, 2) investigated behavioral economics informed interventions, 3) were written in English, 4) were published during the period from 2000 to 2022, 5) used only human samples and 6) had the data needed to calculate the effect size. Studies that did not report odds ratio (OR) were still included in the analysis if OR could be calculated with $2 \times 2$ cross-tabulations. Qualitative studies, gray literature, articles have been published in non-English languages, results have been presented as posters and lectures, and informal reports; were excluded from the analysis.

### Data extraction & literature quality assessment

Two members of the research team independently check extracted the data. Items that were not agreed upon by the two researchers were reviewed and extracted by a third evaluator. From the reviewed studies, information such as the name of the first author, year of publication, country of study, study design, type of intervention, outcome variable and sample size have been extracted. To perform meta-analysis, the total number of cases, the total number of controls, the number of successful cases, the number of unsuccessful cases, the number of successful controls, and the number of failed controls were extracted.

The quality assessment was done by two people so that each person evaluated the studies separately and independently based on the individual participant data the modified Jadad scale (Table B in S1 Appendix).

### Outcomes

The main outcome is the up taking the colorectal cancer screening tests. Studies have used proxies such as: uptake, utilization and intention-to-perform to investigate screening demand. Different types of screening tests are reported in Table 1.

### Statistical analysis

First, the data related to the number of successes and failures in the case and control groups were extracted from the studies as an EXCEL file. The confidence interval of the estimates was considered to be 95%. All the analysis was completed in STATA 15. The pooled effect size has been calculated using fixed and random effects model estimation. In order to check the heterogeneity, we estimated the $I^2$ index. $I^2$ was introduced by Higgins and Thompson. This statistic calculates the variance ratio due to heterogeneity in the estimates of the studies [27]. As well as the drivers of heterogeneity have been identified through meta-regression. Meta-regression estimates the significance of each source of heterogeneity in the pooled effect size through regression analysis [28]. The presence of publication bias was checked graphically by funnel plot, by nonparametric trim-and-fill analysis and by egger test. By performing two analyses, we evaluated the sensitivity of the pooled effect size to important parameters and the role of individual studies. The sensitivity of the pooled effect size to the important parameters and the results of each study was assessed by using leave-one-out meta-analysis. The logic of selecting parameters for sensitivity analysis was the consensus of the authors.

**Table 1.  Characteristics of studies included in the final analysis.**

| Ref | Study | Year | Country | Design | Outcome | Intervention |
|---|---|---|---|---|---|---|
| [17] | Bakr et al | 2020 | USA | RCT | CRC screening uptake | Letter leveraging social psychology and behavioral economics principles |
| [29] | Brenner et al | 2016 | USA | RCT | CRC screening uptake | Decision Aids |
| [30] | Clouston et al | 2014 | Canada | RCT | CRC screening uptake | Patient aid |
| [31] | Gabel et al | 2020 | Denmark | RCT | CRC screening uptake | Self-administered web-based decision aid |
| [19] | Green et al | 2019 | USA | RCT | CRC screening uptake | Mail and lottery |
| [25] | Kullgren et al | 2014 | USA | RCT | CRC screening uptake | Receipt 1 in 10 Chance |
| [25] | Kullgren et al | 2014 | USA | RCT | CRC screening uptake | Raffle for $500 |
| [32] | Lipkus et al | 2005 | USA | RCT | CRC screening uptake | Tailored risk factor information |
| [33] | Lo et al | 2014 | England | RCT | CRC screening uptake | Implementation intentions |
| [22] | Mehta et al | 2018 | USA | RCT | CRC screening uptake | Opt-out of screening |
| [23] | Mehta et al | 2017 | USA | RCT | CRC screening uptake | Active choice |
| [34] | Mehta et al | 2019 | USA | RCT | CRC screening uptake | Lottery incentive |
| [35] | Mehta et al | 2019 | USA | RCT | CRC screening uptake | Active choice |
| [36] | Mehta et al | 2021 | USA | RCT | CRC screening uptake | Loss framed incentive |
| [37] | Mehta et al | 2020 | USA | RCT | CRC screening uptake | Text + Lottery |
| [38] | Menon et al | 2011 | USA | RCT | CRC screening uptake | Tailored counseling |
| [39] | Miller et al | 2018 | USA | RCT | CRC screening uptake | decision aid |
| [40] | Miller et al | 2011 | USA | RCT | CRC screening uptake | web-based decision aid |
| [41] | Myers et al | 2007 | USA | RCT | CRC screening uptake | Tailored message |
| [42] | Myers et al | 2013 | USA | RCT | CRC screening uptake | Preference-based navigation |
| [43] | Myers et al | 2014 | USA | RCT | CRC screening uptake | Tailored Navigation Intervention |
| [44] | Neter et al | 2014 | Israel | RCT | CRC screening uptake | Implementation Intentions |
| [45] | O'Carroll et al | 2015 | USA | RCT | CRC screening uptake | Anticipated regret |
| [46] | Pignone et al | 2011 | USA | RCT | CRC screening uptake | Decision Aid |
| [47] | Schroy et al | 2011 | USA | RCT | CRC screening uptake | Decision aid + personalized risk assessment |
| [48] | Schwartz et al | 2017 | USA | RCT | CRC screening uptake | Nudge |
| [49] | Steckelberg et al | 2011 | Germany | RCT | CRC screening uptake | Evidence based information on risk |
| [50] | Stoffel et al | 2019 | Spain | RCT | CRC screening uptake | Social norms |
| [51] | Stoffel et al | 2021 | Cyprus | RCT | CRC screening uptake | Social responsibility |
| [51] | Stoffel et al | 2021 | Cyprus | RCT | CRC screening uptake | Anticipated regret |
| [51] | Stoffel et al | 2021 | Cyprus | RCT | CRC screening uptake | Account effect |
| [51] | Stoffel et al | 2021 | Cyprus | RCT | CRC screening uptake | Benefit of early detect. |
| [51] | Stoffel et al | 2021 | Cyprus | RCT | CRC screening uptake | Scarcity |
| [51] | Stoffel et al | 2021 | Cyprus | RCT | CRC screening uptake | Social norms |
| [52] | Trevena et al | 2008 | Australia | RCT | CRC screening uptake | Decision aid |
| [53] | Vernon et al | 2011 | USA | RCT | CRC screening uptake | Tailored Interactive Computer-Delivered Intervention |
| [54] | Wardle et al | 2003 | UK | RCT | CRC screening uptake | Psych educational Intervention |

## Results

Fig 1 displays the PRISMA flowchart. In the initial search, 1966 articles were found. After the necessary screenings, 37 studies were included in the analysis.

37 studies were included in the final analysis, in which 72612 cases and 71493 controls have been studied. The Table 1 shows the characteristics of the studies included in the analysis. All studies were randomized clinical trials. As seen in the table, in the reviewed studies, almost all common tests have been investigated as response variables and proxies for colorectal cancer screening demand. In addition, in the aforementioned studies, the effectiveness of various

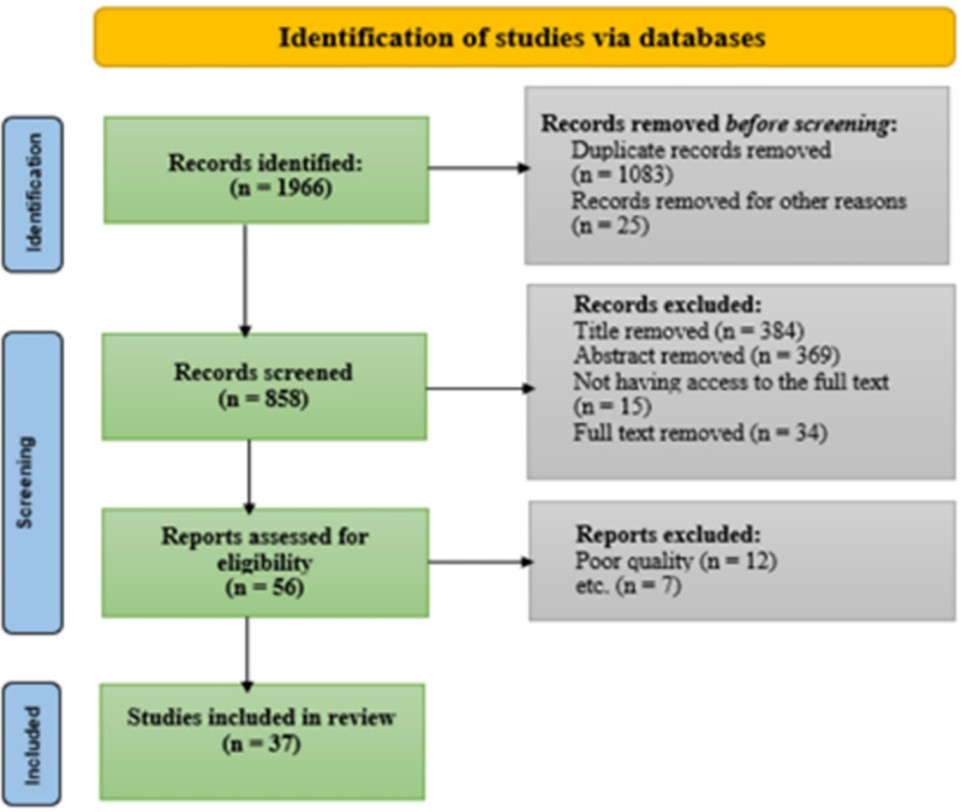

**Fig 1. PRISMA flowchart.**

types of BE informed interventions have been tested. Interestingly, based on our search, most of the studies were conducted after 2010.

Fig 2 shows the frequency distribution of reviewed studies in the world. The red areas show the countries with a high number of studies and the blue areas show the less studied countries, respectively. The areas seen in green are the countries from which we could not find a study. Based on the map, most of the studies found (about 72%) have been conducted in the USA. It is noteworthy that the percentage frequency of studies in other countries is the same (approximately 3%).

## Assessment of heterogeneity size

The obtained information revealed that studies have considerable heterogeneity. So that value of $I^2$ index 94.9% was found (heterogeneity $X^2$ = 731.89, d.f. = 37, p <0.01). Meta regression results in Table 2 determine heterogeneity factors. Differences in follow-up time and quality scores have been the two main sources of heterogeneity between studies (p<0.05). The details of the quality assessment of studies based on The Modified Jadad Scale are shown in Table B in S1 Appendix. It is quite obvious that the effect of these two factors has become insignificant in the random effects model. Therefore, the estimates of the random effects model show us the final effect size.

## Assessment of publication bias

The results of the of publication bias assessment are presented graphically by funnel plot in Fig 3, by nonparametric trim-and-fill analysis in Table 3 and by egger test in Table 4

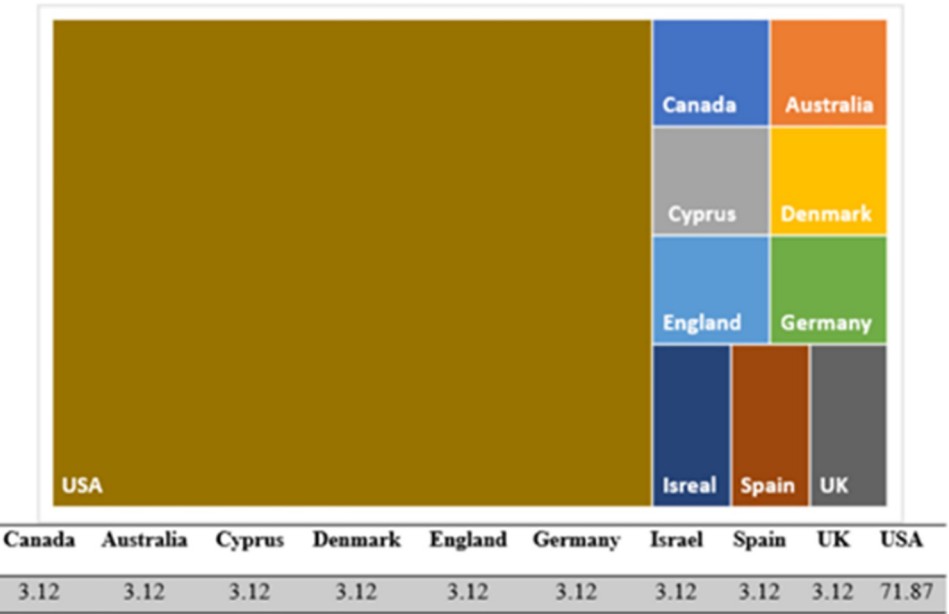

**Fig 2. Frequency of reviewed studies in the world (%), Source: Study findings.**

| Canada | Australia | Cyprus | Denmark | England | Germany | Israel | Spain | UK | USA |
|--------|-----------|--------|---------|---------|---------|--------|-------|-----|-------|
| 3.12 | 3.12 | 3.12 | 3.12 | 3.12 | 3.12 | 3.12 | 3.12 | 3.12 | 71.87 |

respectively. In the funnel plot, the studies are almost asymmetrically distributed, and most of the studies are located at the top of the funnel (that is, studies with high precision). 3 studies are outside the 95% confidence interval. The plot obtained reveals the existence of the publication bias.

Based on plot b in Fig 3 the six studies (orange dots) trimmed and filled on the right side of the funnel plot can be attributed to the possible presence of publication bias. Also, based on the information in Table 3, it is clear that imputing studies (orange dots) on the right side of the funnel plot could lead to an increase in the pooled odds ratio from 1.27 (95% CI: 1.101 to 1.468) to 1.40 (95% CI: 1.209 to 1.624).

As well as Also, the results of the Egger test confirm the existence of publication bias. According to Table 4, it can be easily seen that the bias coefficient is noteworthy (3.48) and statistically significant (p< 0.01).

Considering the substantial size of the heterogeneity, the pooled odds ratio obtained from the random model is acceptable for us. It was estimated as 1.26 (95% CI: 1.11 to 1.43). The Fig 4 illustrations the forest plot by intervention. According to the results presented in the figure, the largest effect size was obtained on opting out intervention 3.86 (95% CI: 2.05 to 7.27), which was related to a single study by Mehta et al. (2018). While the lowest effect size was obtained for Anticipated regret intervention 0.76 (95% CI: 0.52 to 1.11).

**Table 2. Sources of heterogeneity based on meta-regression analysis.**

| Model | Variables | β | SE | [95% conf. interval] |
|-------|-----------|-----|-----|----------------------|
| Fixed Effect | Follow up period | -0.009 | 0.004 | -0.018 to -0.001 |
| | Quality Score | -0.206 | 0.013 | -0.233 to -0.179 |
| | Cons | 1.051 | 0.075 | 0.897 to 1.203 |
| Random Effect | Follow up period | -0.003 | 0.011 | -0.027 to 0.021 |
| | Quality Score | -0.030 | 0.065 | -0.164 to 0.102 |
| | Cons | 0.429 | 0.398 | -0.381 to 1.239 |

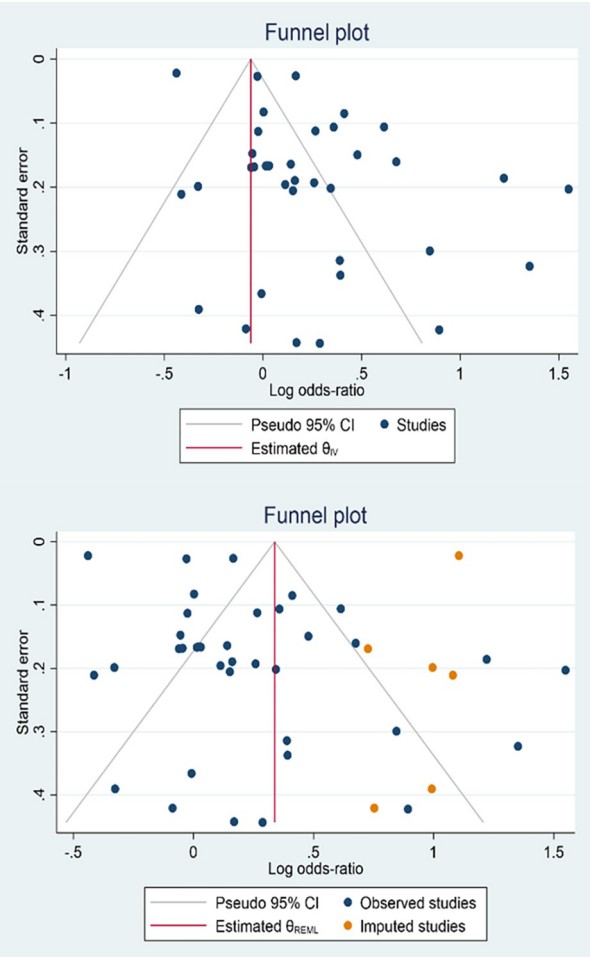

**Fig 3. Checking the existence of publication bias based on funnel plot.** a: Funnel plot based on random effect model, b: Funnel plot based nonparametric trim-and-fill.

**Table 3. Nonparametric trim-and-fill analysis of publication bias, imputing on the right and left.**

| Imputing Side | Studies | Log odds-ratio | [95% conf. interval] |
|---|---|---|---|
| Imputing on the left | Observed | 1.27 | 1.101 to 1.468 |
| | Observed + Imputed | 1.27 | 1.101 to 1.468 |
| Imputing on the right | Observed | 1.27 | 1.101 to 1.468 |
| | Observed + Imputed | 1.40 | 1.209 to 1.624 |

**Table 4. The results of the Egger test (checking the existence of publication bias).**

| Std_Eff | β | SE | t | $P>|t|$ | [95% CI] |
|---|---|---|---|---|---|
| Slope | 0.82 | 0.08 | 10.00 | 0.000 | 0.656 to 0.991 |
| Bias | 3.48 | 1.06 | 3.27 | 0.002 | 1.324 to 5.652 |

Test of H0: no small study effects, p< 0.01

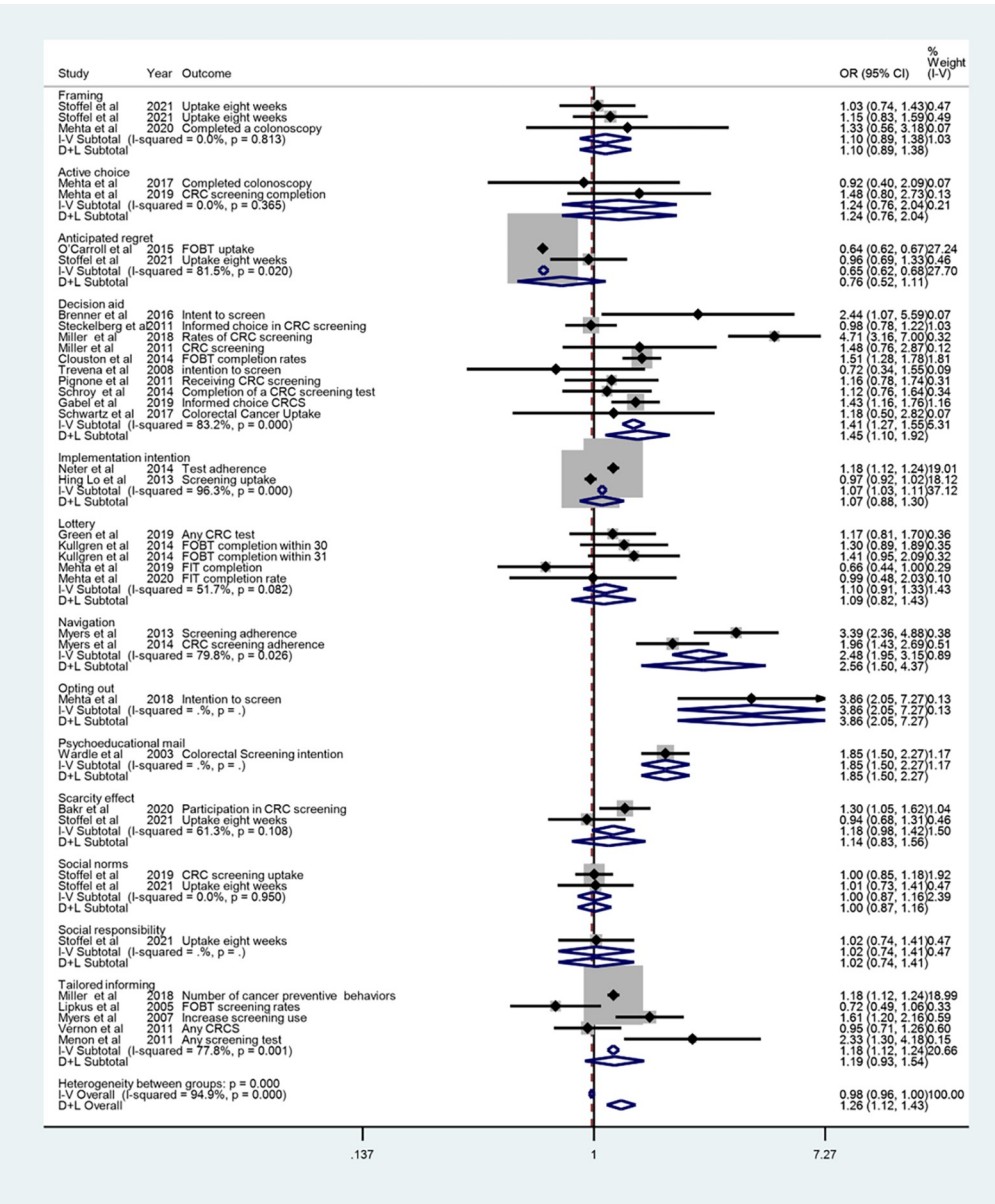

**Fig 4. Results of fixed and random effects meta-analysis by intervention.**

## Sensitivity analysis

The results of sensitivity analysis are presented in Tables 5 and 6. Based on the results of Table 5, the size and significance of the pooled odds ratio has been not sensitive to poor study quality. While after excluding studies that had a follow-up period of less than 3 months and 6 months, the pooled odds ratio had been increased from 1.27 (95% CI: 1.07 to 1.50) to 1.37 (95% CI: 1.16 to 1.61) and 1.35 (95% CI: 1.07 to 1.70), respectively.

Table 6 shows the results of the Leave-one-out meta-analysis. This analysis shows that excluding individual studies causes the pooled odds ratio to fluctuate between 1.22 (95% CI: 1.07 to 1.37) and 1.29 (95% CI: 1.12 to 1.49). That is, by removing and adding each study, the odds ratio changes, but it is still significant.

**Table 5. Results of sensitivity analysis.**

| Sensitivity analysis criteria | Omitted studies | POR | 95% CI |
|---|---|---|---|
| **Quality score < 5** | Wardle et al. (2003), Lipkus et al. (2005) Miller et al. (2011), Pignone et al. (2011) Vernon et al. (2011), Myers et al. (2013) Neter et al. (2014), Clouston et al. (2014) Gabel et al. (2020), Stoffel et al. (2019) Mehta et al. (2019) | 1.27 | (1.07 to 1.50) |
| **Follow up < 3** | Kullgren et al. (2014), Kullgren et al. (2014) Trevena et al. (2008), Stoffel et al. (2019) Stoffel et al. (2021), Stoffel et al. (2021) Stoffel et al. (2021), Stoffel et al. (2021) Stoffel et al. (2021), Stoffel et al. (2021) | 1.37 | (1.16 to 1.61) |
| **Follow up < 6** | Kullgren et al. (2014), Kullgren et al. (2014) Trevena et al. (2008), Stoffel et al. (2019) Stoffel et al. (2021), Stoffel et al. (2021) Stoffel et al. (2021), Stoffel et al. (2021) Stoffel et al. (2021), Stoffel et al. (2021) Gabel et al. (2020), Lo et al. (2014) Mehta et al. (2020), Mehta et al. (2018) Mehta et al. (2017), Mehta et al. (2020) Miller et al. (2011), Wardle et al. (2003) Mehta et al. (2019) | 1.35 | (1.07 to 1.70) |

# Discussion

This study sought to answer this question: Are BE informed interventions effective in increasing demand for colorectal cancer screening? To answer this question, we estimated the pooled effect size of randomized clinical trial studies through a systematic review of published evidence during the period 2000 to 2022. The meta-analysis run in this study covered 144105

**Table 6. Leave-one-out meta-analysis.**

| Omitted study | OR | 95% CI | Omitted study | OR | 95% CI |
|---|---|---|---|---|---|
| **Study** 1 | 1.27 | 1.09 to 1.47 | **Study 20** | 1.25 | 1.08 to 1.44 |
| **Study 2** | 1.25 | 1.09 to 1.44 | **Study 21** | 1.25 | 1.09 to 1.46 |
| **Study 3** | 1.27 | 1.09 to 1.46 | **Study 22** | 1.27 | 1.10 to 1.47 |
| **Study 4** | 1.27 | 1.09 to 1.47 | **Study 23** | 1.29 | 1.12 to 1.49 |
| **Study 5** | 1.27 | 1.10 to 1.47 | **Study 24** | 1.27 | 1.10 to 1.47 |
| **Study 6** | 1.27 | 1.09 to 1.47 | **Study 25** | 1.27 | 1.10 to 1.47 |
| **Study 7** | 1.27 | 1.09 to 1.47 | **Study 26** | 1.27 | 1.10 to 1.47 |
| **Study 8** | 1.29 | 1.11 to 1.49 | **Study 27** | 1.28 | 1.10 to 1.49 |
| **Study 9** | 1.28 | 1.10 to 1.49 | **Study 28** | 1.28 | 1.10 to 1.49 |
| **Study 10** | 1.28 | 1.10 to 1.47 | **Study 29** | 1.27 | 1.10 to 1.47 |
| **Study 11** | 1.27 | 1.09 to 1.46 | **Study 30** | 1.28 | 1.10 to 1.49 |
| **Study 12** | 1.24 | 1.08 to 1.41 | **Study 31** | 1.28 | 1.10 to 1.49 |
| **Study 13** | 1.29 | 1.12 to 1.49 | **Study 32** | 1.28 | 1.10 to 1.49 |
| **Study 14** | 1.28 | 1.10 to 1.47 | **Study 33** | 1.28 | 1.10 to 1.49 |
| **Study 15** | 1.27 | 1.09 to 1.47 | **Study 34** | 1.28 | 1.10 to 1.49 |
| **Study 16** | 1.25 | 1.08 to 1.44 | **Study 35** | 1.28 | 1.11 to 1.49 |
| **Study 17** | 1.22 | 1.07 to 1.37 | **Study 36** | 1.28 | 1.10 to 1.49 |
| **Study 18** | 1.27 | 1.09 to 1.46 | **Study 37** | 1.25 | 1.08 to 1.44 |
| **Study 19** | 1.23 | 1.07 to 1.40 | | | |

participants (72612 cases and 71493 controls). Our estimate of the pooled effect size for all behavioral economics interventions revealed that these interventions have an increasing effect on screening demand. However, the effect size was different based on the type of intervention. In line with our results, Taylor et al. [8] through systematic reviewing the evidence, concluded that the effect of BE intervention on stimulating the demand for screening is significant and that these types of intervention have the potential to improve participation in screening. It is worth mentioning that our study differs from Taylor et al's study in several aspects. First, our search covers a wider range of databases. Second, unlike their study, in addition to the systematic review, we also have estimated the pooled effect size of the interventions. Cadario and Chandon [55] through meta-analyzing of evidence concluded that nudge interventions are more effective in reducing unhealthy eating. By reviewing and meta-analyzing more than 200 studies, Mertens et al. [56] determined choice architecture interventions improve behavior to a small to moderate extent. In another meta-analysis, Beshears and Kosowksy [57] found that choice architecture interventions had a moderate effect. Jachimowicz et al. [58] also reported a moderate effect size for choice defaults. Using a systematic review, Blaga et al. [59] exhibited there is mixed evidence about the effectiveness of behavioral economics interventions in controlling the risk factors of these types of diseases. The results of a systematic review by Valérie and colleagues [60] confirmed the effectiveness of nudging in selecting fruits and vegetables. In a scoping review of the use of behavioral economics interventions in pharmaceutical policy making, the researchers concluded that the effectiveness of the interventions was generally positive, but it depended on the context [61]. Our by group meta-analysis also clarified interesting results. Largest effect size was obtained on opting out interventions. There is a wealth of evidence on the effectiveness of opt out strategies in various behavioral domains. For example, Henriquez-Camacho et al. [62] by systematically reviewing and meta-analyzing 28 articles on the HIV screening uptake, concluded that opt-out strategy was superior to the opt-in, and the refusal rate was lower in the opt-out group. The results of a study indicated that opt-out mailed FIT kit outreach could considerably improve colorectal cancer screening rates in poor population [16]. Asgary et al. [63] also showed opt-out navigation improves breast cancer screening and can reduce multilevel barriers to screening among women with housing problems. A study in Canada found that with the introduction of the opt-out method, HIV testing rates increased [64]. Richter et al. [65] revealed the opt-out strategy folded treatment involvement and increased quit efforts. A study exhibited that opt-out protocol resulted in harmless discontinuation of antibiotics in patients with suspected sepsis [66]. The opt-out strategy takes advantage of the status quo bias. Opting out of a default option has a cognitive cost for people. Based on our estimates, navigation has the second highest effect size. Navigation is an evidence-based intervention that can significantly improve demand for preventive health services [67]. In fact, patients are guided through navigators. Patient navigators can be qualified personnel who help patients overcome changeable barriers to care and attain their care goals [68]. Patient navigation is used in diseases such as cancer screening [69], diabetes [70] and smoking cessation[71, 72]. Navigation intervention can be in the form of educating disease knowledge, education on health system, removing barriers to the medical care, coverage insurance, relaxing other financial barriers, coordinating care, refer to community centers and provide emotional support [68]. By reviewing 67 articles about chronic diseases, Mcbrien et al. [68] concluded that navigation programs improve care for patients. Ritvo et al. [24] showed that the intervention group (navigated) demanded colorectal screening significantly more than the control group. Another part of our findings revealed that leveraging social norms had no effect on stimulating demand for colorectal screening. Social norms indicate acceptable standards of behavior in a society. Since health behaviors are the result of the simultaneous effect of several factors. Therefore, it is possible that the mere leveraging of social norms has a stimulating

effect on behavior such as screening. Unfortunately, we could not find systematic and sufficient evidence about the effect of using social norms in screening. In a cross-sectional study among German men, Sieverding et al. [73] concluded that social norms play an important role in the demand for cancer screening. The results of a study showed that people receiving messages containing social norms performed well in the field of self-monitoring [74]. Notably, we found very little evidence of ineffectiveness of behavioral economics interventions. The results of a field experiment by Bronchetti et al. [75], disclosed that saving behavior did not change among treatment arm (opting out). There is still much debate among researchers and policymakers about the effectiveness of behavioral economics interventions. Some nudges used to change behavior may not be as effective as intended. Nudges that have a large short-term effect may not be very effective in the long term and must be repeated to have a lasting effect. Nudges that effectively change one targeted behavior may simultaneously affect other behaviors. Also, on larger scales, nudges may have tiny effect on planned outcomes. For example, Beshears et al. [76] found that the effectiveness of implementing automatic enrollment on the amount of retirement savings in the first four years of employment was significant but little. Nudges can also have adverse consequences. For example, violation of independence, doubts about positive welfare effects, long-term adverse effects and distortion of democracy and deliberation [77]. The effect size of BE interventions has been reported differently in studies. The range of effect sizes was wide from 0.64 to 3.86. So that, the largest effect size (3.86, 95% CI: 2.05 to 7.23) was reported by Mehta et al. [22]. This study examined the effect of opt-out versus opt-in messaging on fecal immunochemical test (FIT) completion. The authors concluded that opt-out messaging methods can improvement participation in population health outreach works. Owing to the short-term follow-up and the non-comprehensive nature of the studied subjects, the researchers could not perform a downstream diagnostic evaluation of the positive results, so caution should be exercised in generalizing their results in the long term and to other populations. The lowest effect size belonged to the study of O'Carroll et al. [45]. The researchers aimed to measure the effect of a regret evasion intervention on increasing the uptake of colorectal cancer screening. They found the role of regret evasion effective in increasing FOBT uptake. The effect size of this study should also be interpreted with caution. Because in the sample size used, there was little data related to ethnicity. To some extent, this discrepancy could be due to heterogeneity between studies. This heterogeneity can be caused by the variety of interventions and the variety of screening measures. The pooled odds ratio in our study has been estimated more than one and was statistically significant. It means behavioral economics-based interventions are effective in encouraging the demand for colorectal cancer screening. In another review, the authors found compelling evidence that nudges are effective in promoting adherence to guidelines [78]. Möllenkamp and colleague [79] review showed that nudges can improve chronic disease self-management. Summary evidence from a meta-analysis review has revealed that if the information presented about preventive behaviors emphases on the benefits of these actions rather than the losses, people are more likely to perform preventive behaviors [80]. The message frame promoting cancer screening can influence the screening decision [81]. For example, underserved women answered more positively to tailored letter than to a general letter encouraging breast cancer screening. The results of a study by Liang et al. [82] disclosed that nudging (an approach based on behavioral economics) improved the number of appointments scheduled for Medicare health visits and Pap smear test. The authors further conclude that framing and modifying the language of e-messaging can have a significant and long-term impact on patient engagement and access to care [82]. The results of a randomized clinical trial on chronic kidney disease screening in Japan showed that behavioral economics interventions had an improvement effect [83]. Quinn et al. [84] also concluded in their study that the choice architecture based on behavioral economics can

improve the choice of healthy foods by students. In one study, the authors concluded that compared to standard profit-based incentives, offering lottery incentives and losses framed incentives did not lead to a significant increase in HIV testing. However, when low-cost incentives are offered to promote HIV testing, designing lottery prizes may be a better approach than profit-based incentives [85]. Nudges can also have adverse consequences. For example, violation of independence, doubts about positive welfare effects, long-term adverse effects and distortion of democracy and deliberation [77]. The presence of evidence about the minor effects of nudges does not diminish the role of psychological factors in changing behavior. Also, the existence of observations indicating the mild effects of nudges does not mean that behavioral economics techniques such as choice architecture are not useful for policy makers. Behavioral economics is a relatively new area of economics, it does not (and should not) completely replace conventional economics. Rather, it can be considered a complement to neoclassical economics.

## Conclusion

Our meta-analysis revealed that BE interventions are effective in increasing demand for and uptake of colorectal cancer screening. Based on the average cost, BE interventions can be very valuable tools for behavior change [86]. Therefore, health policy and decision makers can improve the efficiency and cost effectiveness of policies to control this type of cancer by using various behavioral economics interventions.

## Limitations

Some limitations can be considered for our study: First, we only analyzed interventions that were directly or indirectly related to behavioral economics. Therefore, some studies that were not in the scope of our keywords may not have been included in the analysis. Second, it was not possible for us to access some specialized databases. Third, we did not include gray evidence and published in non-English languages. Therefore, we suggest that readers generalize our results with caution. Future studies could obtain more accurate estimates of effect size by including a wider range of keywords, searching more comprehensive databases, and examining gray and published literature in other languages.

## Supporting information

**S1 Checklist. PRISMA 2020 for abstracts checklist.**
(DOCX)

**S2 Checklist. PRISMA 2020 checklist.**
(DOCX)

**S1 Appendix.**
(DOCX)

## Author Contributions

**Conceptualization:** Bahman Ahadinezhad, Aisa Maleki, Amirali Akhondi.

**Data curation:** Bahman Ahadinezhad, Aisa Maleki, Amirali Akhondi.

**Formal analysis:** Bahman Ahadinezhad, Aisa Maleki, Amirali Akhondi.

**Investigation:** Aisa Maleki, Sama Yousefy, Fatemeh Rezaei.

**Methodology:** Omid Khosravizadeh.

**Writing – original draft:** Mohammadjavad Kazemi, Sama Yousefy, Fatemeh Rezaei, Omid Khosravizadeh.

**Writing – review & editing:** Bahman Ahadinezhad, Aisa Maleki, Amirali Akhondi, Mohammadjavad Kazemi, Omid Khosravizadeh.

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
