## [Decision Letter · Decision Letter 0]

13 Feb 2023

PONE-D-22-29189Are behavioral economics interventions effective in increasing colorectal cancer screening uptake: a systematic review of evidence and meta-analysis?PLOS ONE

Dear Dr. Khosravizadeh,

Thank you for submitting your manuscript to PLOS ONE. After careful consideration, we feel that it has merit but does not fully meet PLOS ONE’s publication criteria as it currently stands. Therefore, we invite you to submit a revised version of the manuscript that addresses the points raised during the review process.

We look forward to receiving your revised manuscript.

Kind regards,

Ronald Chow

Academic Editor

PLOS ONE

Journal Requirements:

2. We note that Figure 2 in your submission contain [map/satellite] images which may be copyrighted. All PLOS content is published under the Creative Commons Attribution License (CC BY 4.0), which means that the manuscript, images, and Supporting Information files will be freely available online, and any third party is permitted to access, download, copy, distribute, and use these materials in any way, even commercially, with proper attribution. For these reasons, we cannot publish previously copyrighted maps or satellite images created using proprietary data, such as Google software (Google Maps, Street View, and Earth). For more information, see our copyright guidelines: http://journals.plos.org/plosone/s/licenses-and-copyright.

Reviewers' comments:

Reviewer's Responses to Questions

**Comments to the Author**

1. Is the manuscript technically sound, and do the data support the conclusions?

Reviewer #1: Yes

Reviewer #2: Yes

2. Has the statistical analysis been performed appropriately and rigorously? 

Reviewer #1: I Don't Know

Reviewer #2: N/A

3. Have the authors made all data underlying the findings in their manuscript fully available?

Reviewer #1: Yes

Reviewer #2: Yes

4. Is the manuscript presented in an intelligible fashion and written in standard English?

Reviewer #1: Yes

Reviewer #2: Yes

5. Review Comments to the Author

Reviewer #1: Thank you for the opportunity to read manuscript “Are behavioral economics interventions effective in increasing colorectal cancer screening uptake: a systematic review of evidence and meta-analysis?”. This paper addresses and important topic and fills gaps in the literature. I have some comments for authors’ consideration.

Abstract

- Please provide full term of BE

Introduction

- Paragraph 1: please correct typo “098842”

- The authors stated that “various interventions have been investigated to improve the uptake and use of colorectal screening”, what were their results?

- Knowing from the discussion that “Taylor et al” conducted a similar SR recently. What is the difference between yours and Taylor one? What knowledge gaps does your SR intend to address?

Methods

- Considering that BE involves different types of interventions, why was there no subgroup analysis and could not be performed?

Discussion

- You have one long paragraph. This covers multiple topics. Please separate so that there is one topic per paragraph.

Limitation and conclusion

- Please include a limitation in a separate paragraph.

- Any suggestions for future SR?

Reviewer #2: Abstract

• The abstract should summarize key points of the review.

• Use clearer language to describe the main outcome.

• The I2 value presented here does not match the results section.

Introduction

• The text reference ‘Farley et al.’ should be converted to number citation.

• The introduction could use more support by referencing additional studies on behavioural economics. You can begin by providing a definition and presenting examples. Some of the text in the discussion section could be moved here to provide support.

• The introduction can be broken up into paragraphs addressing colorectal cancer statistics, types of screening methods, screening statistics, definitions for terms and concepts such as behavioural economics, how this relates to colorectal cancer screening and the current gap in literature.

Methods

Search Strategy

• Table 1 showing the detailed search strategy should be moved to the appendix.

• Figure 1 showing the PRISMA chart and the accompanying text should be moved to the Results section.

Eligibility Criteria

• What about the exclusion criteria?

• Were there studies with cohort overlap? If so, how was the inclusion/exclusion of these studies handled?

Data extraction

• How many investigators extracted the data? If only one, provide justification. If multiple, how were conflicts addressed?

• ‘Data extraction’ and ‘Literature quality assessment’ sections could be collapsed.

Outcomes

• Use clearer language to describe the main outcome

Data analysis process

• Text on heterogeneity should be removed from this section as it is described in a succeeding section.

Assessment of heterogeneity

• This section could be written using clearer language.

• The heterogeneity described in the results section differs from the one described in the Abstract. Please reconcile.

• ‘Data analysis process’, ‘Assessment of heterogeneity’ , ‘Assessment of publication bias’ and ‘Sensitivity analysis’ should be collapsed into one section ‘Statistical Analysis’.

• Provide a brief description of the tests and plots used for the meta-analysis.

Results

• Use clearer language to describe the results.

• Additional description of the included studies (eg. Proportion of men/women in the studies, participant age range or mean age, number of cases and controls, period of follow-up) could provide context to the potential similarities/differences observed across the studies.

Table 2

• Use consistent terms for outcomes described. eg. 'CRC screening uptake' and 'Uptake' can be unified

Quality Assessment

• Reference should be made to the quality assessment results (presented in the appendix) before presenting analysis results.

Heterogeneity

• The I2 value presented in results does not match what is stated in the abstract. Please reconcile.

Table 3

• Include leading zeros in decimals <1.

Galbraith plot

• The data in the Galbraith plot can also be presented in a table within the appendix to identify the point associated with a particular study.

• How do the three studies that are outside of the 95% limits differ from the rest of the included studies?

Publication bias

• There are more than 3 studies outside the 95% limits in the funnel plot shown in Figure 4a. However, the text above states only 3 studies.

• Result from Egger’s test described in the text (3.15) differs from that in Table 5. Please reconcile.

Funnel Plot

• The forest plot is shown in Figure 5. Firgure 3 is wrongly referenced when interpreting results of the forest plot.

• The figure caption for the forest plot also requires correction.

• Table 6 is not necessary. It conveys the same message as the forest plot.

Sensitivity analysis

• First sentence (“By performing…”) should be moved to the methods section.

Table 7

• It is unclear how/why the three parameters shown in the table were chosen. Please specify in the methods section.

Table 8

• Instead of using ‘Study #’, reference the omitted studies.

Discussion

• Line 7 to 19: Range of effect size should be described in the results section. A brief reference to this range can later be used to discuss potential heterogeneity.

• While multiple studies that used behavioural economics are described in the discussion, it is unclear how these relate to the current systematic review/meta-analysis.

• Discuss whether your findings are reinforced by previous studies or whether they are contradictory? Provide potential explanations.

• Discuss the implications of the findings for public health promotion/policy.

• Strengths and limitations of the review and meta-analysis should be discussed here, not in the conclusion section.

Overall

• Use clearer language throughout.

• Describe abbreviations prior to using them in text. And once described, there no need to repeat the description throughout the text.

• Figure captions and table title should be clear and concise.

• For tables, include a footnote to explain abbreviations/symbols used.

• In the results, indicate which studies are of interest for a specific section by referencing them.

o Eg. When interpreting the Galbraith plots, you can provide references for the two studies that were on the line of “no effect”.

• Break text in the introduction and discussion into paragraphs for readability.

6. PLOS authors have the option to publish the peer review history of their article (what does this mean?). If published, this will include your full peer review and any attached files.

Reviewer #1: No

Reviewer #2: No

---

## [Author Response · Author response to Decision Letter 0]

23 Jul 2023

Dear reviewers,

Thank you for your wise comments.

All your points was corrected and highlighted. 

Best regards

Omid Khosravizadeh

---

## [Editor Report · Decision Letter 1]

9 Aug 2023

Are behavioral economics interventions effective in increasing colorectal cancer screening uptake: a systematic review of evidence and meta-analysis?

PONE-D-22-29189R1

Dear Dr. Khosravizadeh,

We’re pleased to inform you that your manuscript has been judged scientifically suitable for publication and will be formally accepted for publication once it meets all outstanding technical requirements.

Kind regards,

Ronald Chow

Academic Editor

PLOS ONE

---

## [Editor Report · Acceptance letter]

14 Aug 2023

PONE-D-22-29189R1 

Are behavioral economics interventions effective in increasing colorectal cancer screening uptake: a systematic review of evidence and meta-analysis? 

Dear Dr. Khosravizadeh:

I'm pleased to inform you that your manuscript has been deemed suitable for publication in PLOS ONE. Congratulations! Your manuscript is now with our production department. 

Kind regards, 

on behalf of

Mr. Ronald Chow 

Academic Editor

PLOS ONE